# Artificial Intelligence Program for Predicting Wrestlers’ Sports Performances

**DOI:** 10.3390/sports11100196

**Published:** 2023-10-08

**Authors:** Roman Sergeevich Nagovitsyn, Roza Alexeevna Valeeva, Liliia Agzamovna Latypova

**Affiliations:** 1Faculty of Pedagogical and Art Education, Glazov State Pedagogical Institute, 427621 Glazov, Russia; 2Department of Methodology and Technology of Universal Competencies, Kazan State Institute of Culture, 420059 Kazan, Russia; 3Institute of Psychology and Education, Kazan Federal University, 420008 Kazan, Russia; valeykin@yandex.ru; 4Institute of Management, Economics and Finance, Kazan Federal University, 420008 Kazan, Russia; melilek@yandex.ru

**Keywords:** sports performance, artificial intelligence, wrestlers, prediction, program

## Abstract

To date, there are conflicting opinions about the effectiveness of the introduction of artificial intelligence technologies in sports. In this regard, the purpose of the study was to develop and integrate an intellectual program for predicting competitive success into the process of selecting wrestlers to increase its effectiveness. The authors developed a program for predicting the sports performance of wrestlers on the basis of artificial intelligence technology. To implement the study, the individual data of Greco-Roman wrestlers (n = 72) were collected and processed on 36 comparison traits, ranked into categories according to three key areas: sports space, hereditary data and individual achievements. As a result of data processing through means of deep neural networks and machine learning algorithms, two prediction categories were identified: athletes who performed at the sport rank or the highest standard and athletes who did not achieve this standard. Control testing of the created program showed only 11% of error probability in predicting a given wrestler’s competitive performance. As for the functionality of the program in the area of classification of the features by category, the authors’ artificial intelligence program with 100% probability identified key categories of traits that reliably affect the results of the future sports performance of a young wrestler. Thus, the use of neural networks and machine learning algorithms, according to the results of the study, improves the quality of sports selection, which will allow further timely individualization and improvement of the training process of young wrestlers.

## 1. Introduction

Information technology has penetrated into the everyday life of young people, students and society as a whole [1]. Every year, the practical implementation of the achievements of scientific and technological progress in the field of digitalization begins to more significantly affect any of the spheres of human social activity [2]. The introduction of various information programs, neural networks and artificial intelligence (AI) technologies in all spheres of society has reached a stage where the general social system can no longer fully exist without this integration [3]. The implementation of information innovations, including the most relevant of them such as AI programs and machine learning (ML) programs, significantly increase the effectiveness of continuous the social modernization conditions for the quality of human life [4].

Big data, ML and AI, providing the faster development of technology toward human-centric applications, are beginning to globally and fundamentally change the way society and technology develop [5]. With the advent of the Internet of Things, through the use of data-driven conceptual approaches, there are more opportunities to activate new channels for the dissemination of innovations locally and globally in the economy, politics, medicine, education and in all areas of human life [5]. Based on AI technologies, it has become possible to universally implement smart and lean manufacturing, as an integral part of Industry 4.0., in six main aspects of the production line: quality checks, maintenance, reliable design, environmental impact reduction, factory integration and after-production support [6]. Various intellectual platforms are being systematically created through the processes of modeling and marketing simulation to create an analytical ecosystem as one of the main tools for positive social dynamics [7]. Based on the instant processing of big data and their further use, great prospects open up for stimulating innovation and improving public administration, which leads to an increase in the quality of life of the population [7].

As research shows, for many centuries, data mining has been used in various spheres of human activity to process the results of the observations and measurements of real objects, as well as in natural and computational experiments [8,9]. Thus, there is an interesting study on the creation of effective medical services through big data analysis, which proves that multimodal intelligent forecasting can reliably assess the applicability and effectiveness of healthcare policies [10]. On the basis of AI, practical variations of medical services for disease prevention and integrated health policy are ranked through a combination with specific segments of the population, taking into account social problems and spatial and temporal properties [10]. With the help of intellectual technologies, a detailed psychological analysis of college students is implemented when performing various types of physical activity [8]. To date, it has become possible to analyze the indicators of physical health and educational characteristics of students at various stages of sports training based on AI [9]. In the system of physical education and sport as one of the key spheres of society, which has recently been supported by information technologies—and in some cases even artificial intelligence—it is crucial to realize the role of these technologies in increasing the effectiveness of the training process [11,12]. The active use of these innovative technologies increases the reliability of the collection of system data, their accuracy and interpretability for its subsequent analysis to develop an information management system for assessing the physical fitness of athletes [11]. The use of intelligent algorithms improves the quality of refereeing in sports matches and other sports, through an increase in the percentage of objectivity, accuracy, speed of scoring and determining results, as well as timely detection of violations of the rules of athletes [12]. As program and regulatory documents show, the active, practical implementation of intelligent technologies contributes to the creation of effective conditions for improving the quality of life of the population through the intellectualization of services in the sphere of physical education, fitness and sport [13,14]. Additionally, it promotes efficiency of the system analysis of training indicators for professional orientation optimization and early detection of children with outstanding abilities, automation of knowledge quality assessment and analysis of information on the results of the training process [4,15].

In this direction, it is AI that is positioned as a worthy innovative tool able to provide large-scale personalized physical education and training in sports based on big data [16,17]. With the help of artificial neural networks, based on the identification of key performance indicators, it is possible to determine playing talents in professional football as well as their status as field players in the football league [16]. Models are being developed for predicting the quantitative values of physiological indicators of the work of an athlete’s body at the level of anaerobic metabolism threshold when performing ergo-spirometric load testing to failure using ML algorithms [17]. The use of AI helps in the field of sports medicine, namely in the creation of predictive models in the healthcare industry to assess the risk and predict the occurrence of various sports injuries, maintain health stability and physical performance through the analysis of data on internal and external factors [18]. To analyze the technical training of combatants, new intelligent optimization algorithms are being actively used, based on evolutionary calculations through the identification of biomechanical dependencies in martial arts data and the complex neuromuscular control of athletes [19]. Some current empirical studies have proven that the use of intelligent technologies can partially replace coaches and teachers in various organizations of sports profiles and improve the quality of physical education and sports training [20,21]. AI programs help track players, evaluate the effectiveness of each tactical action, create the optimal ratio of sports risk and potential gain in confrontation by calculating useful indicators without the use of painstaking human labeling [20]. Based on the identification and comparison of physiological and positional data during a sports game in the context of a real match, machine learning technologies reliably recognize subsequent technical actions, which makes it possible to correct the managerial decisions of the coaching staff and increase the team’s performance [21].

In the specialized literature, the use of AI technologies on the analysis of big data in the field of sports activities is justified as one of the significant and effective tools to improve the quality of health and motor performance of novice and professional athletes [3,8,10]. It offers solutions to many problems, such as the differentiation of engaged persons to identify dependencies and relationships between them, analysis of the quality of sports training and the risk of trainees receiving poor physical development, automatic construction of recommendations on the use of information resources and materials to more effectively master the sports program [15,22,23].

In the last decade, innovative intelligent technologies have significantly increased their impact on the field of sports, fitness and physical education [16,24]. Intelligent technologies are being used to analyze big sports data, in the aspect of statistics of athletes’ and referees’ actions from match videos or physical activity metrics of competitors [4,14,25]. Based on the use of special sensors and the processing of the received data via machine learning with the help of the neural networks of a deep closed recurrent unit, medical and biological monitoring of the training process is carried out by comparing the electrocardiogram data of athletes and stress, represented by the features of their heart rate variability, which further allow us to model features of psychological stress involved in various sports conditions and implement preventive procedures [26]. AI technologies provide virtual advisory assistance, effectively promoting in the support of athletes’ physical and sports potential and in the realization of cyber sports and sports betting [15,20]. In the pre-competitive stage of sports training, athletes, together with the coaching staff, begin to actively use the results of intellectual analysis to improve the effectiveness of preparation for performance at competitions, as well as AI technologies that allow fully automating the athlete’s pre-start training [10,14].

Indeed, there are conflicting views in the specialized literature on the effectiveness of AI in sports [1,24]; the introduction of these technologies without the implementation of reliable experimental studies may be negative, with the consequence that the next generation will be ill-prepared for a dynamic and changing world with the increasing development of big data [21,23]. However, the purpose of this research is not to introduce global standards based on the introduction of intelligent technologies and certainly not to find ways to replace trainers [4]. The authors aim to create an AI program that, when implemented, will enable coaches and the administration of sports organizations to reliably improve the effectiveness of sports training based on big data [8].

Thus, a preliminary overview analysis of the existing possibilities for the implementation of AI technologies to improve the efficiency of sports training of athletes shows that at the moment there is a special need for the implementation of such an experiment. In this regard, the authors set the goal of the study: to experimentally implement a program for predicting the sports results of athletes in the context of the practical implementation of AI technologies.

Taking into account the issues discussed above, the hypothesis of the study is the following—the implementation of sports selection for sports performance in wrestling will be more effective if an intelligent program for predicting competitive success is developed and implemented to help in the individualization of the training process.

This program would be based on the use of various ML algorithms and big data analysis. Its practical implementation in a sports organization will make it possible to implement the recommendation to identify successful young athletes who have certain types of predispositions to obtain the highest results. The implementation of this program will make it possible to make recommendations for the sports career of a young athlete at the stage of further choosing his sports trajectory, then again at the stage of his training in sports and recreation groups [23]. In this regard, the use of AI in the analysis of big data will help to correct the sports trajectory of a novice athlete [15] and, in the future, will also provide an opportunity for administrative and coaching staff to adjust and personalize sports training at the stage of sports improvement.

## 2. Materials and Methods

During the experimental study (February–September, 2022), the collection and processing of personal data of the wrestlers (n = 72) engaged in Greco-Roman wrestling or who have already completed their wrestling career were carried out. To collect the information, the authors analyzed the archived data (1988–2008) of the wrestlers from the cities of Glazov (the Udmurt Republic, Russia) and Kazan (the Republic of Tatarstan, Russia) and conducted surveys on social networks and via phone. For the study, they collected data from the athletes in the time period when they were just starting their sports career and were young wrestlers. All data were ranked in three directions (“Hereditary Data”, “Sports Space” and “Individual Achievements”) and were collected in a common dataset. Figure 1 below shows a snapshot of the collected datasets of 12 wrestlers for “Sports Space” and “Individual Achievements”.

The focus group for the experiment included wrestlers who currently have the highest sports category “Candidate for Master of Sports of Russia” or the sports title “Master of Sports of Russia” (n = 19), athletes in III–I sports categories (n = 21) and wrestlers who have junior sports categories or have no categories (n = 32). In the experiment, the data of only those athletes (aged 27 to 47 years today) who voluntarily agreed to take part in the experiment and provided the necessary archival information were analyzed.

The study included the development of an intellectual program for creating forecasts of the competitive success of athletes using the Orange analytics system. A workflow for data analysis through means of process intelligence models was created in the interface of the Orange software. Classical deep networks and special machine learning algorithms for categorical classification, “Logistic Regression” and “Random Forest”, were used to implement the experimental workflow [4]. The analytical system Orange was chosen for the study—as this is an open source program for ML, statistical research and data visualization—which has a large set of research and analytical functions. Compared with Megaputer PolyAnalyst, TIBCO Data Science, Polymatica and Anaconda analog intelligent platforms, the Orange Data Mining software uses visual programming, which is implemented via a user-friendly graphical interface with a variety of free analytical functionality and classification algorithms for convenient operation by users without special technical education.

## 3. Results

The predictive and classification functionality of the Orange intelligent platform was used in the analysis of the big data of wrestlers from the focus group. To “train” the author’s program on the Orange smart platform, not all wrestlers’ data were used but only randomly selected data (n = 36) for pre-testing.

As a result of the study, the system analysis of the program created at the previous stage to predict the competitive performance of young wrestlers in the intelligent system Orange allowed the determination of the basic classification for the prediction of sports performance. The program was finally tested with a validation sample (n = 18) selected randomly but with the mandatory condition of sampling uniformity. Using the data from the validation sample, the program independently identified the prognosis on the basis of special intelligent algorithms and neural networks, taking into account the classification of regularities. When an error was detected, additional “training” of the intelligent system was performed until the moment when the created program could independently find the optimal numerical patterns to predict the correct sports result [4]. In conclusion, the effectiveness of the created intellectual program was finally tested using the data of the wrestlers of the control sample (n = 18), which was randomly formed, but on the condition that it included wrestlers with titles, categories, and without sports wrestling qualifications. The data of the wrestlers of the control sample, analyzed with the intellectual program created in the experiment, made it possible to identify the resulting percentage of reliability in the correctness of the forecast proposed by the author’s development under the conditions of using AI.

On the basis of the preliminary analysis of the specialized scientific literature and the peculiarities of young athletes’ versatile monitoring, a system of traits and their categories for the intelligent prediction was developed [1,4,9,12,24,25]. The system was classified into 36 traits, each of which was ranked into 2–4 categories, in three main personal areas: 13 traits via hereditary data, 12 traits via sports space, and 11 traits via individual achievement (Table 1).

After determining the comparative traits by category, the personal data of Greco-Roman wrestlers participating in the experiment, obtained at a time when they were just starting their sports career and participated in their first competitions, were loaded onto the Orange analytical system. As a result of a comparative analysis of the received data of wrestlers based on AI technology, two main levels of sports success prediction were eventually ranked: Greco-Roman wrestlers who had fulfilled the sport rank or achieved the highest level and the wrestlers who had not reached this level. After additional “training” of the intellectual system with validation sampling, the program was tested with data from a control sample (n = 18). As a result of intellectual data processing, only two errors out of the 18 tested data points of wrestlers of the control test sample were reliably recorded, which ultimately indicates an 11% probability of an incorrect answer by the author’s program in the implementation of forecasts of the competitive success of wrestlers. And it should also be specially noted that during the implementation of the intellectual program, only one error was detected out of the 10 athletes when testing the control sample for a group of 10 athletes who did not achieve a high sports result in Greco-Roman wrestling.

Thus, the author’s program for predicting the competitive performance of athletes based on intellectual technologies has been experimentally tested and is ready for practical implementation. Nevertheless, during its practical implementation in the process of sports training of Greco-Roman wrestlers, additional data were required on which traits and categories had a greater impact on performance or, conversely, on the irrelevance for further success in competitive activity. Accordingly, after alternating the features and functionality of the intelligent program, we managed to determine the trait categories most reliably influencing the result “Fulfilling the highest standards for the Candidate for Master of Sports of Russia or the Master of Sports of Russia” (Figure 2).

Figure 2 shows the process of classification of traits and categories in the implementation of the authors’ developed program on the Orange platform. On the left side of Figure 2, the introduction of 36 traits is marked by categorical classification + 1 trait, as an achieved result (will or will not fulfill). This field also denotes the experimental samples (n = 72) that were analyzed. The middle part of Figure 2 shows the workflow of predicting and classifying the loaded data on the intelligent platform by selecting widgets and connections between them. The right margin shows the result of traits classification by category, ranked above in Table 1. As it can be seen from Figure 2, the program determined the key values according to the result “Fulfilling the highest standards for the Candidate or Master of Sports of Russia” in the following sequence in descending order: SS6(2), SS4(2), SS8(2), SS9(2), SS10(2), SS3(2), IA3(2), …. The detected categories of traits indicate their most reliable significance in predicting the wrestlers’ competitive performances.

In turn, with further changing of the intelligent program target setting, there were identified categories of traits that most reliably influence the result “Not fulfilling the highest standards of Candidate for Master of Sports of Russia or the title of Master of Sports of Russia” (Figure 3).

Figure 3 shows the same process of the classification of traits and categories in the implementation of the authors’ program on the Orange platform. Figure 3 demonstrates that the program determined the key values for the opposite result of “Not fulfilling the highest standards of Candidate for Master of Sports of Russia or the title of Master of Sports of Russia” in the following sequence in descending order: SS12(1), SS11(2), IA1(0), IA2(0), IA3(0), SS2(0), SS9(0), …. The identified trait categories indicate their most reliable significance in predicting the competitive performance of wrestlers by the designated negative result for the given study.

After identifying the trait categories that most reliably influence the result of predicting the competitive performance of young wrestlers by analyzing the data from the entire experimental sample (six categories for the positive and six for the negative), the percentage ratio in different combinations of trait categories was obtained (Table 2).

Prediction efficiency in both categories is 100% fixed if in monitoring a young athlete shows three categories of traits together on one of the different options out of every two for each trait. Thus, the program has 100% probability of detecting a reliable prediction when finding three traits in a young wrestler.

In turn, if a young wrestler’s mother is engaged in sports and has an athletic title or one of his elder brothers or sisters has a sports title (one trait with a category in the performance), there is more than 85% probability that he will be successful in future competitive activities in wrestling. And if he is also being coached by a mentor with work experience from 16 to 30 years or if the results of his previous trainees in the last 5 years are not less than Master of Sports of Russia or Master of Sports of International class (categories 1 and 2), then the probability of achieving the highest sports category and Russian rank increases up to 92%.

On the other hand, if a young wrestler was “signed up” by his grandmother or grandfather or was studying in a sports section in a non-sports lyceum (the first indication with the category of not achieving sports results), then he has more than 82% probability to not achieve a high sports level in his career in the future. And if he misses more than 10% of training sessions or his grade point average at school for the previous year before he was engaged in the wrestling section is less than 4.2, then the probability of not achieving the highest sports category and the title of Russia increases to 89%.

## 4. Discussion

The problem of sports selection at the early stages of young athletes’ sports training, as the results of the present study showed, implies the search for regularities, and in the given case, the highest combination of traits by categories compared to the combination of indicators of the whole studied group [4]. It, in turn, comes to the task of ordering system features comparing the indicators of young wrestlers and output results in the form of predictive models based on big data analysis [15,25]. This technology is based on determining the so-called taxonomic distance, that is, the distance between points in the multidimensional space from the identified traits via hereditary data, sport space and individual achievements of young wrestlers [3,10]. As the results of the study demonstrated, the creation of an intelligent system of patterns using comparison traits reached 100%, when at least three traits in the categories coincided. And at the coincidence of only one category of a trait, the probability became 85% on achieving the performance of the highest category of the Candidate for the Master of Sports of Russia or a rank of the Master of Sports of Russia and 82% on not achieving the performance of the highest category or a sports rank.

In the modern period of sports and physical education development, many mathematical methods are used to predict future sports performance at different stages of sports improvement, which is associated with a number of technical and energy-consuming problems in analyzing big data [3,8,15]. The low level of reliability in identifying a meaningful sports selection strategy has a number of important features in creating predictions [16,21], namely, insufficient identification of individual hereditary characteristics and achievements of young athletes, identification and adjustment of traits and categories included in the development, lack of timely risk prediction and limitations on practical applicability [23,25]. In turn, the implementation of AI programs can increase the reliability of prediction detection and significantly improve the accuracy in the processing of athletes’ data when entering the sports section [7,20]. The use of various ML algorithms and neural networks will make it possible to more accurately identify trainees who are not predisposed to a particular sport, which will further open the trajectory of educational impact and individualization of the training process for this group of athletes.

Despite the fact that the use of various ML algorithms and neural networks makes it possible to implement recommendations for identifying athletes who are not predisposed to a wrestling career, this study has a number of significant limitations. In the study, more emphasis was placed on the sections of data on athletes’ “Sports Space” and “Individual Achievements” and less in the direction of “Hereditary Data”, which could significantly affect the effectiveness of the forecast. In the experiment, only the data of athletes in an individual sport, in particular in Greco-Roman wrestling, were taken, which also has significant limitations for using the recommendations of the program in other sports, particularly in team sports. Nevertheless, the methodological recommendations proposed by the developed program can further open a trajectory for the educational impact and individualization of the training process for this group of athletes.

## 5. Conclusions

Thus, in the presented experiment, an attempt was made to look into the future and introduce the intellectual technology of big data into the process of sports training of wrestlers, which is fully consistent with the development of modern martial arts. The developed authors’ program on the basis of AI allows the prediction of the competitive performance of a novice wrestler and reliably captures the main significant categories of traits that positively or negatively affect the prospect of a novice Greco-Roman wrestler achieving the highest sports grade or title of Russia in the future. The practical significance of the study on the implementation of neural networks and ML algorithms was confirmed with the reliable experimental data. Accordingly, the author’s development will improve the quality of the sports selection of young wrestlers and also, in a recommendatory aspect, contribute to the timely personalization and adjustment of the training process of young athletes. In conclusion, it is the artificial intelligence technology with properly selected algorithms for the processes of intellectual clustering, classification and forecasting that has a strong and stable ability for reliable sports analysis, effective selection of the main criteria and indicators that are significant for improving the efficiency of a young wrestler’s sports trajectory.

The results of the study will be of interest to coaches of children’s youth sports schools and schools of higher sportsmanship in martial arts. Practical recommendations for loading and processing wrestlers’ data into the program developed in the study will allow methodologists of sports organizations to timely implement the sports selection of beginner athletes and systematically adjust the training process at the early stages of wrestlers’ training. The peculiarities of the study by stages and the results and conclusions obtained during the implementation of the experiment were repeatedly discussed at methodological meetings and round tables with representatives of sports organizations in the field of Greco-Roman wrestling. In the future, information about the results of this experiment will be disseminated as part of a special module at advanced training courses for wrestling coaches in youth sports schools.

The next study of the authors in the field of the implementation of AI in sports will continue the experiment to supplement big data from other types of martial arts: freestyle wrestling, judo and taekwondo. The experiment will continue to correct the program using additional intelligent algorithms, new systems of neural networks, implementation of a special n-fold cross-validation for higher reliability of the results and the introduction of new features for classifying the processed data. In the future, the main focus of data collection will be focused on the search for the biomedical, psychological and physiological data of young athletes in order to create more accurate predictions of their martial arts sports career.

## Figures and Tables

**Figure 1 sports-11-00196-f001:**
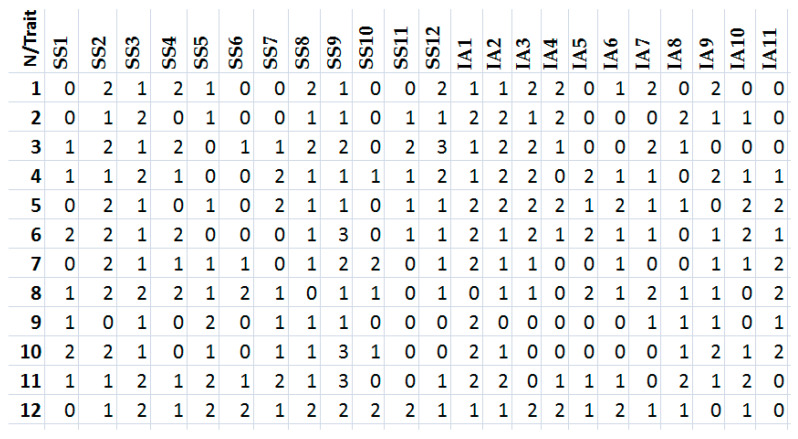
Snapshot of collected datasets of 12 wrestlers for “Sports Space” and “Individual Achievements”.

**Figure 2 sports-11-00196-f002:**
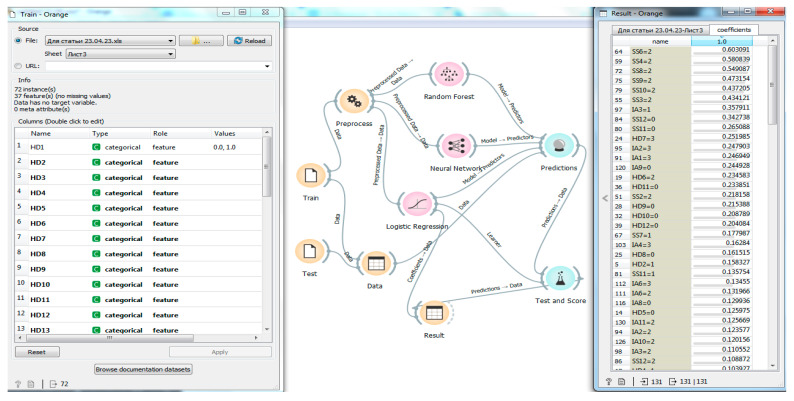
The working window of the developed program on the Orange platform to classify the traits according to the result “Fulfilling the highest standards of the Candidate for Master of Sports of Russia or the Master of Sports of Russia”.

**Figure 3 sports-11-00196-f003:**
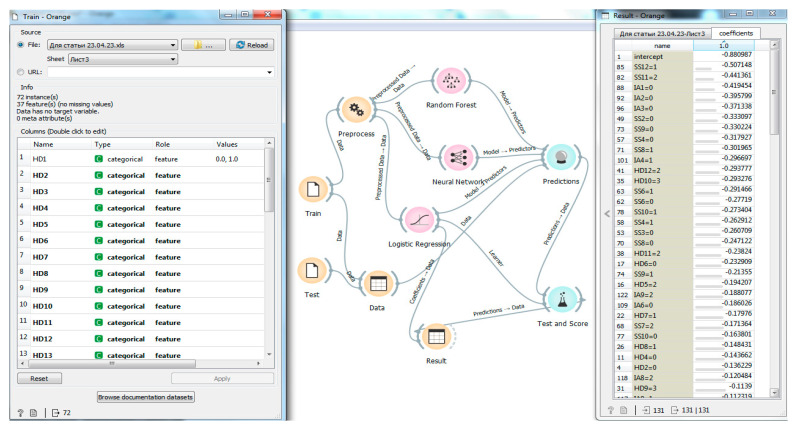
The working window of the developed program on the Orange platform to classify the traits according to the result “Not fulfilling the highest standards of Candidate for Master of Sports of Russia or the title of Master of Sports of Russia”.

**Table 1 sports-11-00196-t001:** Classification of young wrestlers’ data via traits and their categories.

Designation	Trait	Categories by Code Numbers
** *Hereditary Data* **		
HD1	Sex	Young men (1), young lady (0)
HD2	Zodiac sign by the elements	Fire (0), earth (1), air (2), water (3)
HD3	Decade of the zodiac sign	20–28/29/30/31 (0), 1–10 (1), 11–19 (2)
HD4	Maximum oxygen intake	<25 (0), 25–35 (1), >35 (2)
HD5	Blood group	1 (0), 2 (1), 3 (2), 4 (3)
HD6	Hemoglobin	<115 (0), 115–150 (1), >150 (2)
HD7	Hematocrit	<35 (0), 35–45 (1), >45 (2)
HD8	Minerals (%)	<7 (0), 7–10 (1), >10 (2)
HD9	Total fluid (%)	<40 (0), 40–50 (1), >50 (2)
HD10	Protein (%)	<10 (0), 10–15 (1), >15 (2)
HD11	Muscle fibers white/red	<w (0), w = r (1), <r (2)
HD12	Lung vital capacity (mL)	<1500 (0), 1500–2000 (1), >2000 (2)
HD13	Respiratory volume (mL)	<180 (0), 180–250 (1), >250 (2)
** *Sports Space* **		
SS1	Age at the time of “enrollment” in the sports section (year)	7–8 (0), 9–10 (1), 11–12 (2)
SS2	Family status	Orphan (0), single–parent family (1), nuclear family (2)
SS3	Place of residence	Village (0), town with the population of less than 100 thousand residents (1), town with the population of more than 100 thousand residents (2)
SS4	Siblings (elder brothers or sisters)	No siblings (father/mother) or does not practice any sports (0), has an athletic category (1), has a sports title (2)
SS5	Father	
SS6	Mother	
SS7	Coach qualifications	Has an athletic category (0), has the title of the Candidate for Master of Sports (CMS) or the athletic title of the Master of Sports (MS) (1), has the title of the International Master of Sports (IMS) or the Merited Master of Sports (2)
SS8	Results of the coach trainees over the period of the last 5 years	Athletic categories (0), athletic title of CMS (1), athletic title of MS or IMS (2)
SS9	Coach experience	<5 years (0), 5–15 years (1), 16–30 years (2), >30 years (3)
SS10	Father or mother works in physical education and sports	No (0), yes (1), part-time (2)
SS11	Type of educational institution	School (0), gymnasium (1), lyceum (2)
SS12	Who signed up for the sports section	Him- or herself or a friend (0), grandparent (1), brother or sister (2), father or mother (3)
** *Individual Achievements* **		
IA1	Average score at school for the previous year	<4.2 (0), 4.2–4.7 (1), >4.7 (2)
IA2	Ratio of trainings missed to the number of all classes (%)	>10 (0), 5–10 (1), <5 (2)
IA3	Physical development in accordance with the “Ready for Labor and Defense” test	No pin badge (0), bronze/silver pin badge (1), gold pin badge (2)
IA4	Average performance in the first two or three competitions (place)	<3 (0), 2–3 (1), >2 (2)
IA5	Cooper’s strength test (min)	<1.2 (0), 1.2–2 (1), >2 (2)
IA6	Muscle mass index to height and weight	<15 (0), 15–20 (1), >20 (2)
IA7	Fat content (%)	<25 (0), 25–35 (1), >35 (2)
IA8	Fat-free mass (%)	<75 (0), 75-65 (1), >65 (2)
IA9	Muscle mass in relation of the lower limbs to the upper limbs	<1.2 (0), 1.2–1.5 (1), >1.5 (2)
IA10	Muscle mass in relation to right and left extremities	<1.05 (0), 1.05–1.1 (1), >1.1 (2)
IA11	Muscle mass in relation of limbs to body	<1,2 (0), 1.3–1.5 (1), >1.5 (2)

Designation—number encoding of each of the characteristics of the ranking of these wrestlers; Trait—the main directions of wrestlers’ characteristics in the three areas of “Hereditary Data”, “Sports Space” and “Individual Achievements”; Categories by code numbers—categories and indicators of traits in coding for data processing in the Orange system.

**Table 2 sports-11-00196-t002:** Results of reliability in % of categories of traits, credibly affecting the prediction of competitive performance of young wrestlers.

**Categories of traits on the result “Fulfilling the highest standards of Candidate Master of Sports of Russia or the title of Master of Sports of Russia”**		1 *	1–2 **	1–3 ***
1	Athletic mother: has an athletic rank **SS6(2)**/ elder brother or sister: has sports rank **SS4(2)**	>85		
2	Efficiency of trainees in the last 5 years: title of Master of Sports /International Master of Sports **SS8(2)**/trainer’s work experience: from 16 to 30 years **SS9(2)**		>92	
3	Father or mother works in the field of physical education and sports: part-time **SS10(2)**/place of residence: city of more than 100,000 citizens **SS3(2)**			100
**Categories of traits according to the result “Not fulfilling the highest standards of Candidate Master of Sports of Russia or the title of Master of Sports of Russia”**		1	1–2	1–3
1	Who “enrolled” a young athlete in the sports section: grandmother or grandfather **SS12(1)**/ training of a young athlete: lyceum **SS11(2)**	>82		
2	Previous year’s high school grade point average: <4.2 **IA1(0)**/training absences to the number of all classes (%): >10 **IA2(0)**		>89	
3	Physical development on the “Ready for Labor and Defense” test: no pin badge **IA3(0)**/young athlete’s family status: orphan **SS2(0)**			100

* If a wrestler of the first category of traits (one of the two options) is revealed, the corresponding percentage result of the reliability of the sports result is recorded; ** if a wrestler has the first and second categories of traits (one of the two options), the corresponding percentage result of the reliability of the sports result is recorded; *** if a wrestler has three categories of traits (one of the two options), the corresponding percentage result of the reliability of the sports result is recorded.

## Data Availability

Not Applicable.

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
