# Peer review of "Artificial Intelligence Program for Predicting Wrestlers’ Sports Performances"

_sports, 2023, doi:10.3390/sports11100196_

Round 1

Reviewer 1 Report

This work is attractive and heuristic.

Major:

I think the work is meaningful, but the weak point of the manuscript is that for many of the parameter choices and experimental basis, no reason was given. An example is random selection (36, 18...) —what is the principle of their selection? How can you ensure the reproducibility of the results?

The AI part of the article is very much missing, even though it is emphasized in the "Introduction" and "Methods" sections quite a few times. There are no details about deep networks, logistic regression, random forest, etc., except a brief description. How did you adjust the parameters? What are the final configurations of the model for optimal accuracies? How were the models trained?

Besides, n-fold cross-validation can make your results more credible than randomly arranging the data.

While the manuscript's conclusion appears interesting, it is potentially risky to human society. In extreme cases, it even provides a basis for advocating superiority and inferiority. Thus, it should be more rigorously articulated in terms of ethics. Furthermore, "approved by the Ethics Committee" is a fragile statement. Who is the committee? What is the approval number?

L40: In addition to [5, 6], it is worthwhile to pay attention to data mining for analyzing the duration of human daily and sports motions, for example, the up-to-date literature How Long Are Various Types of Daily Activities?

L47: Besides [9, 10], AI applications in related areas include human activity recognition (HAR). The latest work this year is On a Real Real-Time Wearable Human Activity Recognition System, which can successfully recognize a wide variety of sports movements (jumping, kicking in all directions, squatting, kung fu, v-cut, side-step ......) to provide a reference for real-time injury prevention and assisted rehabilitation systems. For HAR/HBR, it is also recommended to cite https://doi.org/10.3390/s23010125 as a comprehensive reference.

Minors:

Well, we're still in an era where we have to write the vast majority of our scientific contributions in English… Smooth phrasing may not be a request for you, but basic grammar and spelling are worth double-checking. Obvious typos and flaws like:

L36 increasingly significantly increase: the first "increasingly" was apparently forgotten to be deleted during rewriting; "increase" should be "increases".

L47 THE efficiency of

Only two instances are provided here, but there exist more. Please check carefully.

see above.

Author Response

Thank you very much for taking thetime to review this manuscript. The correspondingrevisions and corrections highlighted. Please see the attachment.

Reviewer 2 Report

In this manuscript, the authors have conducted a study related with an AI program for predicting wrestlers’ sports performance. Overall, a very well-studied and conducted research, that should consider the following comments:

1.       In the abstract, please provide some background information prior to explaining what the content of this manuscript is. What are the challenges that drove the authors to conduct this study?

2.       In the introduction, please expand the content of references 5,6,7, 8, 12, 13,14,15,16,17 since they are useful for the reader. Placing just the numbers of some references is a shortcoming here.

3.       What is missing in the introduction is the offerings of AI, ML, data mining, big data analytics and so on, in other domains (e.g., healthcare, smart cities, policy making). You should study and properly refer to the following studies:

a.       Kyriazis, Dimosthenis, et al. "The CrowdHEALTH project and the hollistic health records: Collective wisdom driving public health policies." Acta Informatica Medica 27.5 (2019): 369.

b.       Sahoo, Snehasis, and Cheng-Yao Lo. "Smart manufacturing powered by recent technological advancements: A review." Journal of Manufacturing Systems 64 (2022): 236-250.

c.       Biran, Ofer, et al. "PolicyCLOUD: A prototype of a cloud serverless ecosystem for policy analytics." Data & Policy 4 (2022): e44.

d.       Serrano, Martín, et al. "Concepts and Design Thinking Innovation Addressing the Global Financial Needs: The INFINITECH Way Foundations." (2023).

4.       For artificial intelligence and machine learning, you could use the abbreviations of AI and ML accordingly.

5.       Please add a paragraph at the end of the introduction indicating the structure of the rest of the document.

6.       Regarding section 2, how were the 72 wrestlers chosen? Randomly? What were the selection criteria? How could a wrestler opt in and opt out of this study?

7.       In section 2, could you please provide a snapshot of the collected datasets?

8.       Moreover, in section 2, it would be ideal to have a flow diagram in UML (or similar), indicating the flow of the overall process of the study.

9.       In Section 3, why did you choose the Orange intelligent platform? Have you performed any comparison with other similar platforms that concluded into this one?

10.   Figure 1 is not clear – please enhance it

11.   Same comment for Figure 2

12.   In Sections 4 and 5, you should add the limitations and challenges of your work. How did you go beyond them? Did you make any assumptions?

13.   Moreover, In Section 5, it would be a nice addition to include the potential receivers of this work. How are you targeting them and how are you going to communicate/disseminate your research outcomes?

14.   Finally, please provide somewhere clear the next steps and future plans that emerge out of this research.

Author Response

(The authors gave the same response as above.)

Round 2

Reviewer 1 Report

A good revision.

Reviewer 2 Report

The authors have addressed all of the comments! The manuscript can be accepted in its current form! Congratulations and thank you!